# Purification of Lithium Carbonate from Radioactive Contaminants Using a MnO₂-Based Inorganic Sorbent

Olga Gileva [1,*], Pabitra Aryal [1], JunSeok Choe [1], Yena Kim [1], Yeongduk Kim [1,2], Eunkyung Lee [1], Moo Hyun Lee [1,2], Vitaly Milyutin [3], KeonAh Shin [1] and Hyojin Yeon [1]

1   Center for Underground Physics, Institute for Basic Science (IBS), Daejeon 34126, Republic of Korea; moohyun.lee@gmail.com (M.H.L.)
2   IBS School, University of Science and Technology (UST), Daejeon 34113, Republic of Korea
3   Froumkin's Institute of Physical Chemistry and Electrochemistry of the Russian Academy of Sciences (IPCE RAS), Moscow 119071, Russia
*   Correspondence: gilevaolga@ibs.re.kr

**Abstract:** The possibility of deep radiochemical purification of $Li_2CO_3$ has been examined in the context of the purification program of the AMoRE collaboration. In this experiment, commercial $Li_2CO_3$ was converted into $LiNO_3$. Co-precipitation with inorganic salt-based carriers followed by membrane filtration and sorption using MDM inorganic sorbent methods were tested for the removal of alkaline-earth and transition metals, potassium, magnesium, aluminum, uranium, thorium, and radium. The calcium molybdate-based carrier was the most efficient for removing Th, U, and K. Subsequently, the radium, calcium, and barium contamination was removed with MDM sorbent. After the impurities' removal, the final $Li_2CO_3$ product was synthesized with $NH_4HCO_3$ sludge. The separation factors were derived by means of ICP-MS and HPGe analyses of the initial material and the intermediate and final products. The study showed the optimum conditions of co-precipitation and sorption to reach a high yield and radiopurity of lithium carbonate used for low-radioactive-background experiments. The developed method is an important step toward performing next-generation large-scale (1-ton) neutrino experiments using Li-containing detectors.

**Keywords:** lithium carbonate; lithium nitrate; radiochemical purification; co-precipitation; membrane filtration; sorption; MnO₂-based inorganic sorbent; AMoRE experiment

## 1. Introduction

Lithium has been historically used in various fields of science and technology, for example, in the production of ceramic and glass materials, greases, aluminum, and others. In the pharmaceutical industry, it is used to produce medicines to treat mental disorders [1]. In the nuclear and power industries, lithium is used as a heat transfer for nuclear reactors [2], in the burial of high-activity nuclear wastes [3], and for tritium control and capture in controlled thermonuclear fusion reactors [4]. Lithium and its compounds have recently been among the most sought-after rare metals because of their use in the lithium battery industry [1]. In physics science, lithium targets are practically applied in plasma acceleration technology [5]. In particle physics, lithium carbonate is used as a precursor for producing cryogenic bolometric lithium molybdate (LMO) crystals for detecting neutrinoless double-beta decay (0νDBD) of the ¹⁰⁰Mo isotope. The Mo-100 isotope has a comparatively high natural abundance (9.74%) [6] and a high $Q_{\beta\beta}$ value of 3034.40(17) keV [7], while most natural radiation sources have energies below 2615 keV. However, the search for 0νDBD events is very challenging due to their exceptional rarity; for instance, the ¹⁰⁰Mo isotope has a half-life of greater than $1.1 \times 10^{24}$ yr [8–10]. The AMoRE project is a series of experiments designed to search for 0νDBD events of ¹⁰⁰Mo embedded in molybdate-based bolometric crystals using low-temperature calorimeters. AMoRE-II, the second phase of the experiment, aims to probe the corresponding half-life limit of $T^{0\nu}_{1/2} > 5 \times 10^{26}$ years

with a radioactive background lower than $10^{-4}$ ckky at $3034 \pm 10$ keV [11] using an array of approximately 400 $Li_2{}^{100}MoO_4$ crystals. Many efforts must be made to reach the projected background and sensitivity levels. First, radioactive contamination from $^{40}K$, $^{226}Ra$, $^{238}U$, and $^{228}Th$ naturally existing inside the crystals must be reduced to minimize their contribution to the background level. Bolometers for AMoRE-II have been produced with conventional [12] and low-temperature gradient Czochralski [13] techniques. To ensure the radiopurity and uniformity of the synthesized crystals despite the crystallization technique used, high-purity and low-radioactive initial materials, namely, $^{100}MoO_3$ [14] and $Li_2CO_3$ powders, are required. The current study mainly focuses on $Li_2CO_3$ preparation for the AMoRE project. Taking into account the segregation of radioactive impurities during crystal synthesis [12], the $^{40}K$ level below 100 mBq/kg and Th/U at the level of several mBq/kg in the $Li_2CO_3$ precursor would be acceptable. Then, approximately 150 kg of this ultra-low-radioactive, high-purity (over 99.99%) lithium carbonate powder is required to produce 400 LMO crystals for AMoRE-II.

Commercial lithium carbonate at a purity of up to 99.999% costs over USD 1000/kg. For a product such as this, the certificate of analysis (COA) issued by the producer specifies the ppm content of alkali, alkali-earth, transition, and heavy metals but does not contain data on radioactive contamination. Moreover, the radiopurity of the powder may vary from lot to lot; however, it is almost impossible to obtain lithium carbonate powder from a single lot tested in advance. The HPGe gamma spectrometry screening of various samples of 99.99% and 99.999% lithium carbonate powders showed high contamination with $^{226}Ra$ and $^{40}K$ from a few mBq/kg to Bq/kg [15]. Thus, developing a radiochemical purification method for producing high-purity lithium carbonate powder is a crucial issue for the AMoRE project.

The scope of countless published studies on lithium purification has mainly focused on purification and recovery to meet the ever-increasing demand for battery-grade lithium. Most articles and patents discuss the removal of K, Na, Cs, Mg, B, etc., from lithium chloride, bromide, sulfate, nitrate, hydroxide, bicarbonate, etc. solutions. The current study is devoted to the possibility of removing radioactive contamination to a level of approximately mBq/kg. Additionally, this study aimed to achieve both an ultra-low level of radioactivity and an ultra-high level of purity over 99.999% for lithium carbonate powder. This paper is structured as follows: The materials and methods used are described in Section 2. The results of co-precipitation with different carriers are presented in Section 3. Section 4 describes the sorption purification process using $MnO_2$-based sorbent. Finally, Section 5 provides the conclusion and a discussion of the results.

## 2. Materials and Methods

Lithium nitrate purification was performed in a class 1000 clean room (ISO6) at the Center for Underground Physics (CUP) of the Institute for Basic Science (IBS) in the Republic of Korea [16]. Pharmaceutical-grade lithium carbonate powder from the Novosibirsk Rare Metal Plant (Russia) [17] was selected as the initial material due to its comparatively reasonable price and the possibility of buying all the required 150 kg from the same batch. Nitric acid (67–70%, TraceMetal-grade) was purchased from Fisher Chemical™ and used without any purification for dissolving the initial $Li_2CO_3$ powder. Lithium hydroxide monohydrate from Sigma-Aldrich® (ACS reagent, $\geq$ 98.0%) was used to adjust the initial lithium nitrate solution pH. Nitric acid purified with the Savillex® DST sub-boiling acid purification system and aqueous ammonium hydroxide solution supplied by Sigma-Aldrich® (~25% Puriss) were used in the final steps of purification and for pH adjustment.

A calcium-based carrier for co-precipitation was produced from commercially available calcium carbonate and purified at the CUP. A calcium molybdate ($CaMoO_4$, CMO)-based carrier was synthesized from CUP-purified calcium carbonate and natural molybdenum trioxide powders [18] through the interaction of calcium nitrate and ammonium molybdate solutions. The carriers were introduced into the lithium nitrate solution at

approximately 5 mol%. Following the co-precipitation with the $Li_2CO_3$ carrier, the initial lithium carbonate powder was taken in about 5% access amount to the required $HNO_3$ for complete dissolution. After the carrier was introduced, the pH of the mixture was adjusted with LiOH to 9; then, the mixture was stirred well and left overnight to let the sediment grow and settle down. The sediment was filtered out using a membrane with a 0.1 μm pore size PTFE filter (Advantec®, Tokyo, Japan).

The filtrate obtained after the membrane filtration was subjected to a sorption purification process in the dynamic mode with a manganese (III, IV) oxyhydrate-based MDM sorbent [19]. A semicommercial batch of the MDM sorbent, TU (Technical Specification) 2641-001-51255813-2007, was produced by the Frumkin Institute of Physical Chemistry and Electrochemistry, Moscow, Russia. Prior to being used, the sorbent was pre-cleaned and conditioned to Li-form. The sorbent was soaked in a 1% $HNO_3$ solution to remove small dust particles, rinsed, and charged into Savillex® 120 mL PTFE column. Precleaning was performed by successive treatments with 3 mol $L^{-1}$ $HNO_3$, followed by 0.1 mol $L^{-1}$ $HNO_3$. Lithium nitrate solution with a concentration of 1 mol $L^{-1}$ and a pH of 9 was used for column conditioning. Purification was performed by passing the $LiNO_3$ solution through the column at a rate of 3 bed volumes (b.v.) per hour. The solution was collected in fractions, where K, Ba, Sr, Th, and U were analyzed with ICP-MS at the CUP. The results of this analysis were used to assess the purification efficiency and define the separation factors. The separation factors were determined from the ratio of the initial material's impurity concentrations to those in the purified product.

The $LiNO_3$ solution received after the column purification was evaporated to a concentration of approximately 8 mol $L^{-1}$ and filtered with a membrane with a 0.1 μm pore size PTFE filter (Advantec®, Japan); then, the pH was lowered to approximately 2. The acidified solution was kept in the fridge at 4 °C overnight to force crystallization. The $LiNO_3$ crystallohydrate was filtered under vacuum and dried. To avoid melting the product, the crystals were dried at room temperature for three days under vacuum (<10 mTorr); then, the temperature was slowly raised to 120 °C to ensure water removal. Dry lithium nitrate powder was tested with HR-ICP-MS at the SEASTAR™ [20] and HPGe gamma spectrometry at the CUP [21]. The final lithium carbonate powder was synthesized through an interaction with ammonium bicarbonate sludge (Fisher Chemical™, SLR-grade). Before the final synthesis of $Li_2CO_3$, approximately 5% of the stoichiometric amount of $NH_4HCO_3$ was added to a solution of lithium nitrate saturated at 40 °C and mixed until a small amount of $Li_2CO_3$ sediment was observed. The sediment was filtered with the same membrane filter, and the synthesis was completed. The synthesized lithium carbonate powder was rinsed several times with hot deionized water and dried.

## 3. Lithium Nitrate Purification Via Co-Precipitation

Once received from the company producer, raw lithium carbonate powder was tested to understand the level of chemical and radioactive contamination (Table 1).

Along with the HPGe testing, the content of alkali, alkali-earth, transition, and heavy metals was studied with HR-ICP-MS. Measurements showed unacceptable activities of $^{226}$Ra (from the $^{238}$U decay chain) and $^{228}$Ac/$^{228}$Th (from the $^{232}$Th decay chain), while the $^{40}$K and $^{137}$Cs activity levels were low enough to be used in AMoRE-II crystal synthesis. The Ca, Ba, Mg, B, Fe, and Na concentrations were over the acceptable one ppm level and were required to be reduced to avoid contamination of the AMoRE-II crystals. The lithium nitrate solution was selected as an intermediate compound for further purification due to the simplicity of preparing pure nitric acid and the possibility of applying it to the manganese oxide-based sorbent. Purified lithium nitrate can be converted into carbonate by interacting with $NH_4HCO_3$ sludge or $CO_2$ and $NH_3$ gases [22].

**Table 1.** The concentrations of metallic impurities (HR-ICP-MS) and radioactivity levels (HPGe) measured in the initial lithium carbonate powder.

| Element | Concentration [ppb] | Element | Concentration [ppb] | Radionuclides | Activity [mBq/kg] |
|---|---|---|---|---|---|
| Al | $530 \pm 50$ | Mn | $80 \pm 20$ | $^{40}$K | $\leq 66$ |
| B | $4800 \pm 500$ | Na | $1720 \pm 150$ | $^{137}$Cs | $\leq 3$ |
| Ba | $15,200 \pm 1500$ | Ni | $230 \pm 30$ | $^{234}$Th | $\leq 151$ |
| Ca | $36,000 \pm 5000$ | Pb | $18 \pm 2$ | $^{234m}$Pa | $\leq 162$ |
| Cr | $650 \pm 150$ | Sr | $310 \pm 50$ | $^{226}$Ra | $2730 \pm 140$ |
| Cs | $420 \pm 50$ | $^{232}$Th | $0.09 \pm 0.02$ | $^{228}$Ac | $110 \pm 10$ |
| Cu | $120 \pm 30$ | Ti | $140 \pm 20$ | $^{228}$Th | $9 \pm 3$ |
| Fe | $1100 \pm 200$ | $^{238}$U | $1.1 \pm 0.1$ | | |
| K | $770 \pm 70$ | V | $130 \pm 20$ | | |
| Mg | $19,500 \pm 1500$ | Zn | $300 \pm 50$ | | |

The initial lithium nitrate solution obtained after the raw $Li_2CO_3$ powder was completely dissolved was dark brown, indicating high iron contamination. By increasing the pH of the solution to 9, hydroxides that are insoluble in alkali media precipitated out of solution. Various carriers were introduced into the solution to force the co-precipitation of insoluble impurities. Calcium-based and carbonate-based carriers were selected for Ra removal. Our previous studies demonstrated that the CMO-based carrier [14,18] efficiently separates thorium and radium. After the co-precipitate was filtered out using the PTFE membrane filter, the lithium nitrate solution was analyzed with ICP-MS at the CUP (Table 2).

**Table 2.** Impurity reduction in lithium nitrate using the co-precipitation purification method. The values are normalized with the $LiNO_3$ concentration.

| | K [ppb] | Sr [ppb] | Ba [ppb] | Ca [ppm] | Fe [ppb] | Pb [ppb] | Th [ppt] | U [ppt] |
|---|---|---|---|---|---|---|---|---|
| Initial material | $770 \pm 70$ | $310 \pm 50$ | $15,200 \pm 1500$ | $36 \pm 5$ | $1100 \pm 200$ | $18 \pm 2$ | $90 \pm 20$ | $1100 \pm 100$ |
| Co-prec. with $Li_2CO_3$ | $430 \pm 50$ | $160 \pm 30$ | $1200 \pm 150$ | $30 \pm 5$ | $200 \pm 50$ | $10 \pm 1$ | $\leq 6$ | $200 \pm 30$ |
| Co-prec. with $CaCO_3$ | $270 \pm 30$ | $9350 \pm 500$ | $1400 \pm 150$ | $50 \pm 5$ | $200 \pm 50$ | $0.7 \pm 0.2$ | $\leq 6$ | $150 \pm 20$ |
| Co-prec. with $CaMoO_4$ | $300 \pm 30$ | $1100 \pm 100$ | $900 \pm 100$ | $45 \pm 5$ | $<100$ | $<0.1$ | $\leq 6$ | $\leq 6$ |

All selected carriers were found to efficiently remove the measured contaminants, but calcium molybdate showed the highest decontamination efficiency. Barium, used as a Ra indicator, was reduced by 15 times. The uranium concentration was reduced by three orders of magnitude and the thorium concentration by one order, resulting in the project's satisfactory limit of six ppt. Alkali potassium does not form insoluble compounds under given conditions, and the separation is possible only due to occlusion or mechanical entrapment. Measurements confirmed that the K concentration decreased by two times and was found to be approximately 0.3 ppm. Strontium and calcium concentrations in the resulting solution were affected by cross-contamination from Ca-based carriers themselves but could be smoothed out in the following column purification. Remarkably, the removal of uranium in the presence of carbonate anion was much less efficient than CMO-based co-precipitation, which could be explained by the formation of stable soluble uranyl carbonate complexes [23].

## 4. Lithium Nitrate Sorption Purification

After CMO-based co-precipitation, the lithium nitrate solution was subjected to a sorption purification process using MDM sorbent. The sorbent works as an ion exchanger, and the chemical composition of the sorbent is as follows:

$$(\mathrm{Na,K})_{0.23-0.25}\mathrm{Mn}^{(\mathrm{III})}_{0.23-0.25}\mathrm{Mn}^{(\mathrm{IV})}_{0.75-0.78}\mathrm{O}_2 \cdot (1.6-1.8)\mathrm{H}_2\mathrm{O} \tag{1}$$

The sorbent was converted into $H^+$-form and then conditioned into $Li^+$-form with lithium nitrate. The affinity series $Ra^{2+} > Ba^{2+}$, $Pb^{2+} > Sr^{2+} >> Ca^{2+}$ explains the sorption-selective characteristics. The impurity removal rate was tested for 1 mol $L^{-1}$, 4 mol $L^{-1}$, and 7 mol $L^{-1}$ LiNO$_3$ solution at pH 8. Approximately 50 column volumes of each solution were passed through the pre-cleaned, conditioned sorbent. The first 10 b.v. were collected separately to ensure column validation. The experiment results were summarized in Table 3. It was found that the sorption efficiency significantly decreased as the LiNO$_3$ concentration in the solution increased. The highest separation factors were observed for 1 mol $L^{-1}$ lithium nitrate solution, showing Ba and Ca reduction levels of over three orders of magnitude and a Sr reduction level of over two. The slight decrease in purification efficiency for the 4 mol $L^{-1}$ LiNO$_3$ solution was insignificant considering the overall improvement in the production rate.

**Table 3.** Separation factors (SFs) for MDM-sorbent purification in LiNO$_3$ solution with different molarities.

| Separation Factor | K | Sr | Ca | Ba | Pb | Th | U |
|---|---|---|---|---|---|---|---|
| 1 mol $L^{-1}$ LiNO$_3$ | 0.1 | 210 | 60 | 2100 | >50 | $\geq 1$ | $\geq 1$ |
| 4 mol $L^{-1}$ LiNO$_3$ | 0.1 | 200 | 70 | 1900 | >50 | $\geq 1$ | $\geq 1$ |
| 7 mol $L^{-1}$ LiNO$_3$ | 0.3 | 4 | 10 | 60 | >40 | $\geq 1$ | $\geq 1$ |

Before and after column purification, thorium and uranium concentrations were below the detection limit of 6 ppt, indicating no detectable leaching from the sorbent. Apart from Th and U, the potassium concentration value in the eluate increased by one order of magnitude. Comparatively higher lithium affinity for manganese oxide than $H^+$ and $K^+$ ions [24], along with the high Li capacity of MnO$_2$ [25], explain the elution of potassium from the sorbent. To make the MDM sorbent purification method applicable to low-radioactive background experiments, the conditioning procedure and long-term stability of the sorbent were studied further.

Hence, for further investigation, a 4 mol $L^{-1}$ lithium nitrate solution was selected. After completing the MDM column precleaning and conditioning, as explained in Section 2, approximately 1000 bed volumes of the LiNO$_3$ solution were passed. ICP-MS samples were taken every 100–200 bed volumes to assess the long-term stability of the sorbent and the elution profiles of the sorbates. Barium, strontium, and calcium concentrations were used to indicate column capacity and the long-term resistance of the sorbent to the highly concentrated solution. The results are presented in Figure 1.

Within the 10 bed volume of the column purification, the concentrations of Sr and Ba were reduced by three orders of magnitude, and within 50 b.v., by four orders, resulting in a detection limit of about 150 ppt for both. No breakthroughs were detected until 950 bed volumes of the solution were passed. The calcium concentration was reduced by two orders of magnitude after passing 50 b.v. of the solution and was stable at approximately $600 \pm 50$ ppm until 800 b.v. was passed. From 800 b.v. to 950 b.v., the calcium concentration increased to approximately 700 ppb, indicating a 10% breakthrough.

As in the study of the separation factor as a function of LiNO$_3$ molarity, the K concentration in 50 b.v. increased by one order of magnitude relative to the initial value. Maximum contamination was observed in the first 10 b.v. of the eluate. After 50 b.v., the potassium leached from the sorbent less but continuously until 950 b.v.

The thorium and uranium concentrations in the initial solution were below the detection limit of 6 ppt, but elution isotherms after column purification showed different behaviors. While the thorium level remained at less than six ppt during the procedure, the uranium concentration rose slightly after 200 b.v. and reached 50 ppt at 950 b.v.

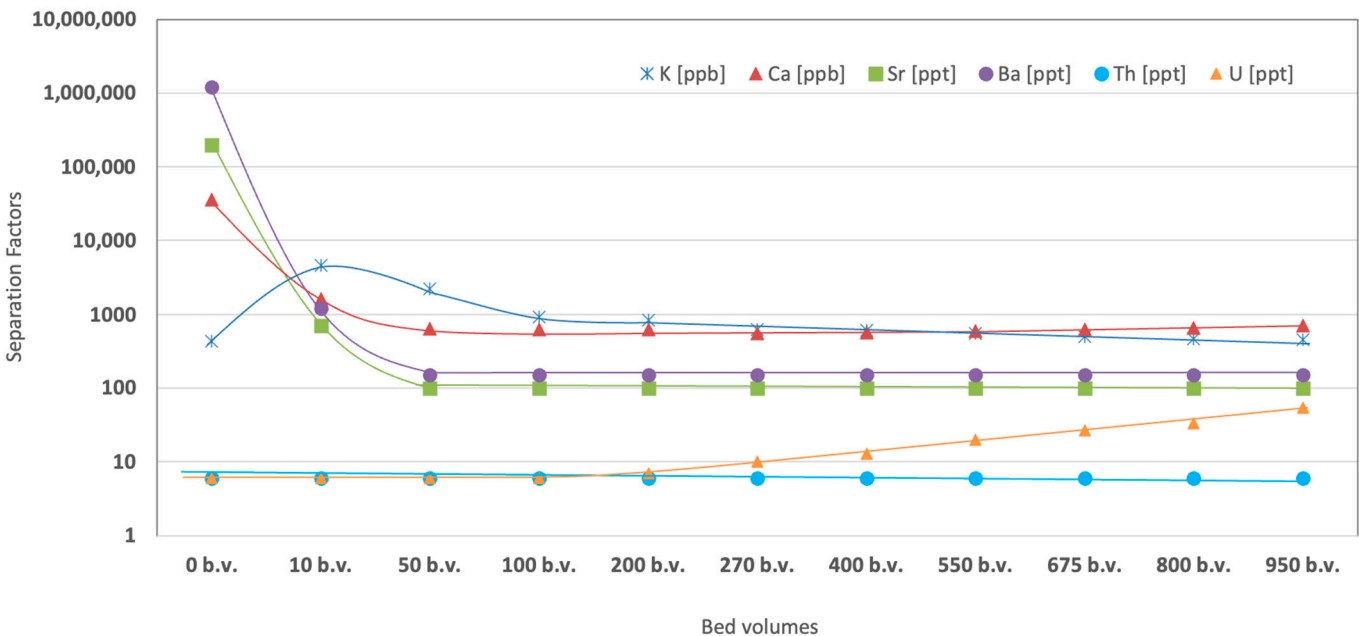

**Figure 1.** Breakthrough curve of K, Ca, Sr, Ba, Th, and U on MDM sorbent in 4 mol L$^{-1}$ LiNO$_3$ at pH 8.

Lithium nitrate crystals were crystallized from the collected purified solution to further investigate the radioactivity reduction with MDM-sorbent purification. The eluate from 50 to 950 b.v. was evaporated twice in a volume and cooled at 4 °C overnight. The obtained crystals (approximately 60% crystallization efficiency) were dried to remove crystalline water and measured with HPGe (Table 4) and HR-ICP-MS (Table 5). Then, the final lithium carbonate powder was synthesized and tested to understand the applicability of the purification method and the possibility of reaching the projected background level requirements for the Li$_2$CO$_3$ precursor.

The contaminants' concentrations and radioactivity levels in purified lithium products were compared with those of the initial Li$_2$CO$_3$ in Figure 2. The reduction in impurities in the purified LiNO$_3$ and final Li$_2$CO$_3$ powders was plotted to show their behaviors at different purification stages.

**Table 4.** The concentrations of metallic impurities (HR-ICP-MS) and radioactivity levels (HPGe) measured in the final lithium nitrate and lithium carbonate powders.

| Element | Concentration [ppb] | | Element | Concentration [ppb] | |
|---|---|---|---|---|---|
| | Purified LiNO$_3$ | Final Li$_2$CO$_3$ | | Purified LiNO$_3$ | Final Li$_2$CO$_3$ |
| Al | 460 ± 50 | ≤300 | Mn | 180 ± 30 | ≤10 |
| B | 110 ± 10 | 30 ± 5 | Na | 150 ± 30 | 50 ± 20 |
| Ba | 1.3 ± 0.3 | 1.0 ± 0.1 | Ni | 70 ± 20 | 40 ± 10 |
| Ca | 600 ± 120 | 260 ± 30 | Pb | 0.20 ± 0.02 | 0.8 ± 0.2 |
| Cr | 60 ± 10 | 50 ± 10 | Sr | ≤0.2 | 1.5 ± 0.5 |
| Cs | ≤10 | ≤10 | $^{232}$Th | ≤0.01 | ≤0.01 |
| Cu | 5 ± 1 | 4 ± 1 | Ti | 5 ± 1 | 3 ± 1 |
| Fe | ≤80 | 220 ± 30 | $^{238}$U | 0.12 ± 0.02 | 0.33 ± 0.03 |
| K | 2700 ± 300 | 300 ± 30 | V | ≤1 | ≤1 |
| Mg | 6.3 ± 1.2 | 1.0 ± 0.2 | Zn | 500 ± 100 | 100 ± 10 |

**Table 5.** The concentrations of metallic impurities (HR-ICP-MS) and radioactivity levels (HPGe) measured in the final lithium nitrate and lithium carbonate powders.

| | Activity [mBq/kg] | |
|---|---|---|
| | Purified $LiNO_3$ | Final $Li_2CO_3$ |
| $^{40}K$ | $77 \pm 6$ | $\leq 14$ |
| $^{137}Cs$ | $1.2 \pm 0.2$ | $\leq 1.0$ |
| $^{234}Th$ | $\leq 16$ | $\leq 22$ |
| $^{234m}Pa$ | $\leq 47$ | $\leq 47$ |
| $^{226}Ra$ | $\leq 1.0$ | $\leq 1.5$ |
| $^{228}Ac$ | $\leq 1.4$ | $\leq 1.4$ |
| $^{228}Th$ | $\leq 1.0$ | $\leq 1.4$ |

According to the reference studies [26,27], in comparison with other types of organic and inorganic sorbents, mixed-valence (III, IV) manganese oxides have a relatively high selectivity for $Sr^{2+}$, $Cs^{137+}$, and $Ra^{2+}$ in the presence of interfering ions of $Mg^{2+}$, $Ca^{2+}$, and $Na^+$. However, these sorbents are usually synthesized using $KMnO_4$ and contain a decent amount of potassium in their structure. The purity of the synthesized sorbent was also determined by the purity of the raw materials. In the studied 4 mol $L^{-1}$ $LiNO_3$ solution containing Ba, Ca, and Mg at the ppm level, the highest separation factors were observed for Ra226 and Ba137. In the purified lithium nitrate powder, radium contamination was reduced by over three orders of magnitude, reaching the detection sensitivity limits. Barium was reduced by a factor of four orders of magnitude. The strontium and calcium concentrations decreased by three and two orders of magnitude, while magnesium showed a separation factor of approximately ten. The ionic radii of $Mg^{2+}$ and $Zn^{2+}$ are very similar to those of $Li^+$ [6]; thus, separating these elements is difficult [28,29]. Indeed, weak zinc leaching from the sorbent was observed in the lithium nitrate sample. Together with zinc, leaching of K and Mn, matrix components of the sorbent, was found at a level of one order of magnitude. The initial cesium contamination was comparatively low. However, ICP-MS and HPGe confirmed a reduction to approximately one mBq/kg. The sodium concentration was reduced from the ppm level to the ppb level, showing similar behavior to that of Cs.

The MDM sorbent purification combined with the co-precipitation method covers a wide range of contaminants. Slight calcium and strontium contamination from the Ca-based co-precipitation carrier was eliminated by further sorbent purification. In two purification steps overall, the boron contamination that often accompanies lithium was eliminated by a factor of 50, while no reduction in aluminum was observed. Other tested heavy and transition metals were reduced by one or two orders of magnitude, resulting in the required sub-ppm level. Based on ICP-MS analysis, thorium and uranium contamination was eliminated at the co-precipitation step, and no further Th contamination was observed during sorbent purification. Apart from thorium, uranium was slightly leached from the sorbent, causing secondary contamination of the solution.

For the final synthesis of lithium molybdate crystals, lithium nitrate could not be used and was converted into lithium carbonate. The technical performance of bubbling $NH_3$ and $CO_2$ is comparatively complicated, so a stoichiometric amount of the sludge of ammonium bicarbonate was used for $Li_2CO_3$ synthesis. Commercial ammonium bicarbonate is usually radiochemically pure because it is produced from gaseous $NH_3$ and $CO_2$. The co-precipitation step with a $Li_2CO_3$-based carrier was also introduced to eliminate the secondary K, Mn, and Zn contamination from the sorbent. The suggested procedure improved the purity of the final $Li_2CO_3$ product. Additionally, Na, K, Mg, Mn, and Zn were reduced by one order of magnitude. At the same time, a slight increase in Fe, Pb, and U concentrations was observed, which could be explained by cross-contamination from ammonium bicarbonate and the preconcentration of impurities on the surface of the freshly produced powder.

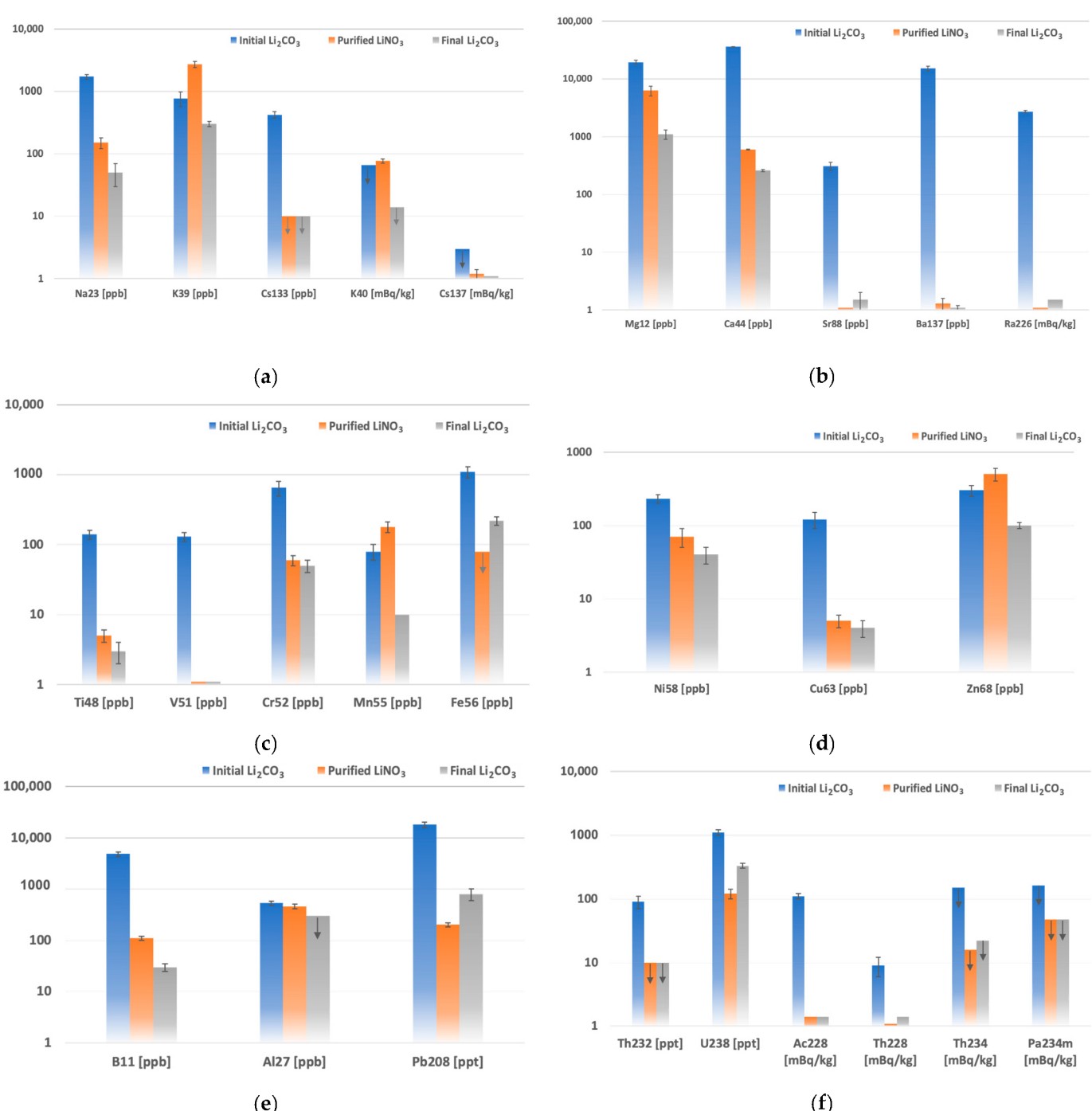

**Figure 2.** Reduction of impurities in purified LiNO$_3$ and Li$_2$CO$_3$ products: (**a**) Removal of $^{23}$Na, $^{39}$K, $^{133}$Cs, $^{40}$K, and $^{137}$Cs; (**b**) Removal of $^{12}$Mg, $^{44}$Ca, $^{88}$Sr, $^{137}$Ba, and $^{226}$Ra; (**c**) Removal of $^{48}$Tl, $^{51}$V, $^{52}$Cr, $^{55}$Mn, and $^{56}$Fe; (**d**) Removal of $^{58}$Ni, $^{63}$Cu, and $^{68}$Zn; (**e**) Removal of $^{11}$B, $^{27}$Al, and $^{208}$Pb; (**f**) Removal of $^{232}$Th, $^{238}$U, and their decay chain products. The arrows indicate the detection sensitivity limit.

Overall, the carbonate-based carriers were less efficient at removing uranium from lithium nitrate solutions. Without interfering with the carbonate anion, uranium, thorium, and potassium were removed efficiently with the calcium molybdate-based carrier. The residues of calcium and strontium in the solution were eliminated with radium in the following MDM sorbent purification process. The LiNO$_3$ concentration of 4 mol L$^{-1}$ was selected for sorption purification at pH 8. Regarding long-term operation, the sorbent was stable, and the calcium 10% breakthrough was observed only after passing 950 bed volumes

of the solution. Potassium, manganese, zinc, and uranium were leached into the solution from the sorbent, causing secondary contamination. The additional step of co-precipitation (with $Li_2CO_3$) before synthesizing the final lithium carbonate product efficiently eliminated the cross-contamination from K, Mn, and Zn. However, uranium was not removed with a small amount of the co-precipitate and was preconcentrated with the final product.

## 5. Conclusions

Purifying lithium carbonate for use in low-radioactive-background experiments is challenging. The expected issue of potassium removal is accompanied by high radium contamination. Considering commercial production technology [30], lime (CaO) is one route in which many impurities can find their way into lithium products, notably radium, which commonly occurs in limestone. Combining co-precipitation with inorganic salt-based carriers and sorption with inorganic sorbents is an efficient strategy for removing a wide range of chemical and radioactive impurities from lithium salts.

All chemical impurities were reduced to the required levels following the suggested purification procedure. Radium was decreased by three orders of magnitude, resulting in a satisfactory level of approximately one mBq/kg. Despite cross-contamination with the sorbent's matrix components, the K concentration was reduced to below the required 0.5 ppm. ICP-MS and HPGe analyses confirmed that potassium was reduced by a factor of two at a minimum. Thorium was easily removed below 10 ppt at the beginning of the process, and no secondary contamination was observed. Relative to the initial commercial powder, uranium activity was reduced by a factor of three in the final product. However, we could not reach the required uranium level due to the difficulty of separating it in the presence of carbonate anion. Efficient in uranium removal, CMO-based carriers could not be used in the final steps to prevent Ca and Sr cross-contamination. Additional purification steps must be implemented to eliminate U contamination from the sorbent, for example, ultrafiltration, using chelating or ion exchange resins, recrystallization of the final $Li_2CO_3$ via the carbonization method, etc. Regarding production efficiency, the MDM sorbent was highly efficient, with over 900 bed volumes of 4 mol $L^{-1}$ $LiNO_3$ solution that could be purified.

The developed method is an important step toward performing next-generation large-scale (1-ton) neutrino experiments using Li-containing detectors and could be useful for lithium production in the nuclear industry [31], Li-ion battery recycling [32], low-radioactive water and seawater treatment, etc.

**Author Contributions:** Conceptualization, methodology, and writing—original draft preparation, O.G. and V.M.; investigation, validation, and technical performance, P.A., Y.K. (Yena Kim), K.S. and H.Y.; ICP-MS analysis, J.C.; HPGe analysis, E.L.; supervision, M.H.L.; project administration and funding acquisition, Y.K. (Yeongduk Kim); writing—review and editing, all authors. All authors have read and agreed to the published version of the manuscript.

**Funding:** This work was supported by the Institute for Basic Science (IBS), funded by the Ministry of Science and ICT, Republic of Korea (Grant id: IBS-R016-D1).

**Data Availability Statement:** The authors confirm that none of the data in the article have been published elsewhere and that all the data are available in the article itself.

**Acknowledgments:** The authors are pleased to acknowledge the support of the SEASTAR™ analytical laboratory in performing HR-ICP-MS analysis of the samples.

**Conflicts of Interest:** The authors declare no conflict of interest.

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
