# Peer review of "Purification of Lithium Carbonate from Radioactive Contaminants Using a MnO2-Based Inorganic Sorbent"

_inorganics, doi:10.3390/inorganics11100410_

Round 1
Reviewer 1 Report
This manuscript discusses the purification of Lithium Carbonate from radioactive contaminants using MnO2–based inorganic sorbent. The article was written as experimental stages. The authors describe how they solved a specific challenge. They have to describe other purification methods that are published in the literature, and they have to apply their method to a few materials apart from Li2CO3. Also, each result has to be explained by the chemistry and physics characterization. The method is based on purification, but no solubility constant is given in order to explain the results.
Some specific comments:
1. At the end of section 1- the introduction, there is a description of the article sections, I think it's unnecessary.
2. The arrangement of Tables 1 and 4 is confusing; table 1 has to be only with 2 columns, and table 4 has to be only with 3.
3. The first time an abbreviation is mentioned, it must be defined.
4. In each figure, bars that describe the error have to be.
Reviewer 2 Report
Gileva et al. in their study investigate the possibility of deep radiochemical purification of Li2CO3 from radioactive waste using MnO2-based inorganic sorbent. The authors used both co-precipitation and sorption procedures. In my point of view, the manuscript is scientifically sound, however, there are some changes that need to be solved before it can be accepted for publication. Here are the details of necessary revision:
1- There are some grammatical mistakes. Minor editing of English language required.
2- Abstract: clarify MDM sorbent.
3- Authors must point out the novelty of their study noticeably.
4- Did you convert lithium nitrate to carbonate then convert it back to nitrate for sorption purification?
5- What is the chemical formula of CMO-based carrier?
6- Table 1. 238U, 232Th
7- Schematic diagram shows the process is needed.
8- Error bars are missing.
9- The conclusions section is too long; it needs to be paraphrased.
Minor editing of English language required
Round 2
Reviewer 1 Report
1. One of the highlights has to be an explanation of the importance of the results.
2. The results' importance must be described in the abstract and the conclusions sections.
Author Response
Dear reviewer,
The results' importance was explained in the Highlights, Abstract, and Conclusion sections.
The newly made changes are marked green.
Please refer to the resubmitted file.